# Effects of Salinity and Temperature on the Flexibility and Function of a Polyextremophilic Enzyme

**DOI:** 10.3390/ijms232415620

**Published:** 2022-12-09

**Authors:** Victoria J. Laye, Shahlo Solieva, Vincent A. Voelz, Shiladitya DasSarma

**Affiliations:** 1Institute of Marine and Environmental Technology, University System of Maryland, Baltimore, MD 21202, USA; 2Department of Microbiology and Immunology, University of Maryland School of Medicine, Baltimore, MD 21202, USA; 3Department of Chemistry, Temple University, Philadelphia, PA 19122, USA

**Keywords:** halophile, psychrophile, enzyme kinetics, molecular dynamics simulations, β-galactosidase

## Abstract

The polyextremophilic β-galactosidase enzyme of the haloarchaeon *Halorubrum lacusprofundi* functions in extremely cold and hypersaline conditions. To better understand the basis of polyextremophilic activity, the enzyme was studied using steady-state kinetics and molecular dynamics at temperatures ranging from 10 °C to 50 °C and salt concentrations from 1 M to 4 M KCl. Kinetic analysis showed that while catalytic efficiency (*k_cat_*/*K*_m_) improves with increasing temperature and salinity, *K*_m_ is reduced with decreasing temperatures and increasing salinity, consistent with improved substrate binding at low temperatures. In contrast, *k_cat_* was similar from 2–4 M KCl across the temperature range, with the calculated enthalpic and entropic components indicating a threshold of 2 M KCl to lower the activation barrier for catalysis. With molecular dynamics simulations, the increase in per-residue root-mean-square fluctuation (RMSF) was observed with higher temperature and salinity, with trends like those seen with the catalytic efficiency, consistent with the enzyme’s function being related to its flexibility. Domain A had the smallest change in flexibility across the conditions tested, suggesting the adaptation to extreme conditions occurs via regions distant to the active site and surface accessible residues. Increased flexibility was most apparent in the distal active sites, indicating their importance in conferring salinity and temperature-dependent effects.

## 1. Introduction

The mechanism of polyextremophilic protein function is of significant interest for pushing the limits of protein evolution and the design of novel enzymes for biotechnology. Of particular interest are proteins from extremely halophilic Archaea (haloarchaea), salt-loving microorganisms in the third Domain of life. Enzymes from haloarchaea typically function optimally in salt concentrations of 2–4 M or higher, reflecting the internal milieu of cells [1]. The halophilic property of these enzymes reflects the common strategy of haloarchaea to balance osmotic stress by “salting-in”, accumulating salts internally to balance the high salt concentrations in the external environment. A combination of genomic and biochemical studies have established that haloarchaeal proteins evolved high surface negative charges to increase their solubility and maintain their structure and function in extremely high salinity [2,3,4].

In the case of enzymes of haloarchaea inhabiting perennially cold environments, both cold temperatures and high salinity must be tolerated simultaneously [5]. An excellent source of such polyextremophilic enzymes is *Halorubrum lacusprofundi*, isolated from Deep Lake in the Vestfold Hills of Antarctica, which grows under both high salinity and low temperatures [6]. The salinity of Deep Lake is 28–32% NaCl (*w*/*v*) with a temperature range of −18 to 11.5 °C. The high concentration of salts results in freezing-point depression which prevents the lake from freezing even in the coldest months. *H. lacusprofundi* has been found to be capable of growth in the laboratory at temperatures as low as −2 °C and its enzymes are consequently adapted to function under such low temperatures and high salinity conditions and has even been of interest for astrobiological studies [7,8].

To better understand the function of polyextremophilic enzymes of *H. lacusprofundi*, the genome was sequenced and the predicted proteome analyzed in relation to cold activity [9,10]. The proteome of *H. lacusprofundi* was found to be highly acidic with a mode of 4.4 for the pI, similar to other sequenced haloarchaea. Its highly acidic proteome is the hallmark of haloarchaeal proteomes, observed originally with sequencing of the *Halobacterium* sp. NRC-1 genome and subsequently confirmed in many other related species [1,2,4]. In order to determine amino acid compositional biases related to cold activity, the *H. lacusprofundi* proteome was compared to twelve haloarchaea from temperate climates [9]. A total of 604 *H. lacusprofundi* proteins belonging to fully conserved haloarchaeal orthologous groups (cHOGs) [11] were aligned to homologous proteins from the twelve other haloarchaea. Amino acid substitutions were observed in 7.85% (5541 residues diverged out of a total 70,589) of the *H. lacusprofundi* cHOG protein positions invariant in homologous proteins from the mesophilic halophiles [9]. The most common substitutions observed were: (1) glutamic acid with aspartic acid or alanine, (2) small non-polar and polar residues with other small polar or non-polar amino acids, and (3) aromatic residues, especially tryptophan, with other aromatic residues [9].

To analyze the mechanism of extremophilic protein function experimentally, the *H. lacusprofundi bga* gene which codes for its cold- and salt-active β-galactosidase was selected for further investigation. The *H. lacusprofundi* β-galactosidase has a high negative net charge located primarily on its surface, typical of haloarchaeal proteins, and its amino acid alterations compared to mesophilic homologs are similar to those observed at the whole proteome level [9,12]. To explore the structure and function of *H. lacusprofundi* β-galactosidase, the enzyme was expressed using *Halobacterium sp*. NRC-1 as host, which lacks an endogenous β-galactosidase enzyme. The enzyme had measurable activity below 0 °C, and surprisingly above 60 °C, with an optimum of 50 °C, and functioned under extremely high salt conditions: 2.0–4.5 M NaCl or KCl, with an optimum of 4.0 M for either salt [12]. The *H. lacusprofundi* β-galactosidase was initially modeled using the published structure of the homologous *Thermus thermophilus* family 42 β-galactosidase (pdb: 1KWK) and the model was subsequently validated by an X-ray crystal structure (pdb: 6LVW) [13,14]. The family 42 β-galactosidases contain a TIM barrel with eight repeats of βα units, with E142 (β-strand 5) and E312 (β-strand 8) predicted to serve as the catalytic residues in *H. lacusprofundi* [9].

In subsequent structure–function analysis of *H. lacusprofundi* β-galactosidase, six residues were found to be diverged in the cold-active enzyme compared to mesophilic haloarchaeal homologs [12,15]. Site-directed mutagenesis and steady-state kinetic studies were conducted which found five of the amino acid residues to be important for enhanced cold activity or reduced activity at moderately high temperatures. A series of double mutations showed that key residues with substantial surface exposure distal from the active site (16–48 Å) were dominant [16]. These results indicated that temperature-dependent kinetic effects were complex and subtle and are likely mediated by changes to the hydration shell and/or perturbation of internal packing.

In the current investigation, we addressed the mechanism of *H. lacusprofundi* β-galactosidase enzyme function with varying temperatures and high-salt conditions through a combination of steady-state kinetics and molecular dynamics simulations. Through this interdisciplinary approach, we correlated kinetic effects under different salt concentrations and temperatures with protein dynamic predictions, to address the fine-tuning required for optimal enzyme activity in two extremes.

## 2. Results

### 2.1. Steady-State Kinetics Reveal Catalytic Rate Constants at Varied Temperatures and High Salt Concentrations

To address the mechanism of modulation of *H. lacusprofundi* β-galactosidase activity at various temperatures and salinity, we initially conducted steady-state kinetic analysis of the wild-type enzyme expressed in *Halobacterium* sp. NRC-1. Using the classic chromogenic substrate o-nitrophenyl-β-galactoside (ONPG), we carried out kinetic measurements from 1–4 M KCl in one-molar increments and temperatures of 10–50 °C in ten-degree increments. The catalytic rate constant (*k*_cat_) and Michaelis constant (*K*_m_) were determined from Lineweaver–Burk plots and used to calculate the catalytic efficiency (*k*_cat_/*K*_m_). The largest relative effect of salinity on catalytic efficiency was observed at the lowest temperature, consistent with the requirement of high salinity for improved low-temperature activity (Figure 1).

Catalytic efficiency showed improvements with both increasing temperature and salinity, with the highest values observed for 4 M KCl at 50 °C (Figure 1a). This is consistent with the previously documented optima of 4.5 M KCl or NaCl and 50 °C [12]. The lowest catalytic efficiency values were at 1 M KCl and 10 °C. However, at 10 °C, increasing the salinity from 1 M KCl to 2 M KCl resulted in a remarkable > 500% increase in the catalytic efficiency. Increasing the salinity from 2 M to 3 M at 10 °C results in an increase, but of only 47%, and from 3 M to 4 M results in a 17% increase. In contrast to 10 °C, at 50 °C, increasing the salinity from 1 M to 2 M results in a smaller 150% increase, 2 M to 3 M results in a 23% increase, and 3 M to 4 M results in a 15% increase (Figure 1a). The relatively large increase in catalytic efficiency observed from 1 M KCl to 2 M KCl at both temperature extremes compared to the smaller increases seen with higher molarities indicates that a threshold of 2 M KCl is needed to reach optimum function for this halophilic enzyme. The catalytic efficiency improves at all salinities as temperature increases but the effect of increasing the salinity is lessened, with the dramatic increase from 1 M to 2 M KCl decreasing in magnitude with increasing temperature, and the most prominent effects at the lowest temperatures.

A driving force in the observed catalytic efficiency changes is the increase in catalytic rate constant, *k*_cat_, which improved with increasing temperature, especially from 1 M to 2 M KCl. Similar to the catalytic efficiency, the observed effects of salinity on the catalytic rate constant reflected the minimum molarity or threshold of salinity required for improved enzyme activity (Figure 1b). The increased salinity effects on the catalytic rate constant also resulted in the largest improvement at 10 °C, similar to the catalytic efficiency. Although the lowest catalytic rate constant observed was at 1 M KCl, increasing salinity to 2 M resulted in dramatic 4- to 5-fold increases at 10 °C and substantially less (~80%) at 50 °C (Figure 1b). At 10 °C, the catalytic rate constants only increase by 5% when the salinity is increased from 2 M to 3 M. The catalytic rate constant maximum is at 3 M KCl, being 17% higher compared to 4 M KCl (Figure 1b). The largest difference between 4 M KCl and 2–3 M KCl is at the intermediate temperature of 20 °C, with a 30–40% higher rate constant at 2 M and 3 M, respectively (Figure 1b). The difference then decreases to a 21% improvement over 4 M for 3 M KCl at 40–50 °C, following the same trend observed in catalytic efficiency (Figure 1b). These findings show that there is a threshold salinity required for optimal catalytic rate, after which there is little further improvement and indeed a slight lessening of the rate at progressively higher salinities.

The observed *K*_m_ values at different salinities and temperatures indicated generally higher temperature effects on the *K_m_* at the higher salinities, with few exceptions. The *K*_m_ plots versus temperature showed enhanced differences for 2–4 M KCl, with 1 M and 2 M trending closely to one another (only 3–5% higher at 1 M) (Figure 1a). An exception was at the highest temperature, 50 °C, where there is a 35% increase from 2 M to 1 M. The *K*_m_ slopes for 3 and 4 M are increased from 10 to 40 °C compared to the lower molarities, a 8–32% increase from 2 M to 3 M, and a 51–123% increase from 2 M to 4 M, consistent with a greater effect of temperature on binding to substrate at higher salt concentrations, as expected for a cold-active enzyme (Figure 1a).

### 2.2. Enzyme Activation Barrier Is Lower with Reduced Enthalpy and Entropy at Higher Salinity 

It has been proposed that psychrophilic enzymes achieve cold adaptation through a reduced temperature dependence of their catalytic rates [17]. According to the Eyring model of chemical kinetics, the catalytic rate *k*_cat_ of an enzyme can be described by
$$k_{cat}=\left(\frac{k_{B}T}{h}\right)e^{-\Delta H^{‡}/RT}e^{+\Delta S^{‡}/R},$$
where *k_B_* is Boltzmann’s constant, *T* is the temperature, *h* is Planck’s constant, *R* is the gas constant, ∆*H*^‡^ is the activation enthalpy, and ∆*S*^‡^ is the activation entropy. Psychrophilic enzymes often have lower activation enthalpies and activation entropies than their mesophilic and thermophilic homologs, resulting in a catalytic rate that is less temperature dependent [18,19,20]. The lower activation entropies of psychrophiles are consistent with increased enzyme flexibility, with a greater cost of preorganization to reach the catalytic transition state.

To examine how the apparent activation enthalpy and entropy of *H. lacusprofundi* β-galactosidase catalysis changes with salt concentration, we used a Bayesian inference method (see Materials and Methods) to estimate the activation enthalpy ∆*H*^‡^ and activation entropy ∆*S*^‡^ at each KCl concentration, from the temperature dependence of the measured *k*_cat_ values (Figure 2). The results indicate an enzyme with ∆*H*^‡^ = 90.4 kJ mol^−1^ and ∆*S*^‡^ = 78.0 J K^−1^ mol^−1^ at 1 M KCl, while at 2–4 M KCl, both the activation enthalpy and entropy are significantly lower (∆*H*^‡^ ≈ 65 kJ mol^−1^ and ∆*S*^‡^ ≈ 10 J K^−1^ mol^−1^). Consequently, this analysis is consistent with *H. lacusprofundi* β-galactosidase exhibiting psychrophilic character only at high salinity (2–4 M KCl) and behaving more like a mesophilic enzyme at 1 M KCl. 

### 2.3. Molecular Dynamics Simulations Reveal Greater Flexibility with Increasing Salt Concentration, but a Rigid Catalytic Core

To better understand the dynamics of *H. lacusprofundi* β-galactosidase in atomic detail, and its role in halophilic and temperature adaptation, we constructed a homology model of the enzyme, and performed molecular simulations at 1 M, 2 M, 3 M, and 4 M KCl (see Materials and Methods). For each salt concentration, five different temperatures were used: 10, 20, 30, 40, 50 °C, resulting in 20 sets of conditions. Parallel simulations were performed on the Folding@home distributed computing platform, to produce a collection of trajectories ranging in length from ~20–80 ns, for a total aggregate simulation time of 422 µs. Each set of conditions contains 494 trajectories. To avoid bias, trajectory lengths were curated to achieve the same distribution of trajectory lengths across all sets of conditions (Appendix A).

The average root-mean-square fluctuation (RMSF) of backbone alpha-carbons observed in the simulations shows striking differences in flexibility for different regions of the protein. Across all conditions, the catalytic core (domain A) containing the active site and TIM barrel has the lowest RMSF values (0.065–0.072 nm at 10 °C, 0.075–0.082 nm at 50 °C) of the three domains. Domain B displays more flexibility (0.087–0.092 nm at 10 °C, 0.10–0.11 nm at 50 °C), and domain C the most flexibility (0.125–0.15 nm at 10 °C, 0.158–0.166 nm at 50 °C). Domain B contains three long and flexible loops, corresponding to residues 485–503, 519–551, and 555–577, which can be identified from the peaks of the RMSF profile (Figure 3). Domain C is even more flexible, especially for residues 640–670 and the last ten residues of the C-terminus. The flexibilities of these regions are consistent with the regions of unresolved residues (531–544 in domain B and 656–673 in domain C) in the recently published crystal structure of *H. lacusprofundi* β-galactosidase (PDB: 6LVW), as well as the results of 100 ns simulations of this structure (Karan et al., 2020).

As the KCl concentration increases, there is a clear increase in flexibility across all domains, with the largest contribution to the average RMSF coming from domain C (Figure 3, inset). While the average per-residue RMSF of catalytic domain A increases from ~0.065 to 0.072 nm at 10 °C as salt concentration increases from 1 M to 4 M, key active site residues in the catalytic core remain relatively unperturbed. Of eight residues surveyed in the active site (R103, N141, E142, Y267, E312, W320, E360, and H363), four had significantly below-average RMSF (~0.040 nm at 10 C), including the E142 and E312, the likely acid/base catalyst, and nucleophile, respectively (Figure 4). The other four residues had average or above-average RMSF values, including R103 and E360, which reach across the TIM barrel from separate loops to coordinate the substrate.

The low RMSF values observed in domain A and the relatively unperturbed RMSF values in the eight catalytic core residues suggested that flexibility of the solvent-exposed residues of the protein might be correlated to solvent exposure. To test this idea, we defined the surface of the protein as the set of residues that contain an atom which exceeds a solvent-accessible surface area (SASA) cutoff value. This SASA cutoff value was set as the 80th percentile of the average SASA per atom values for each set of conditions (Appendix A). The most flexible residues are on the surface of the protein, while the flexibility of the interior of the protein remains more constant throughout the different temperatures, which is expected for globular proteins (Figure 5). The RMSF of solvent-exposed residues increases by 5.6% from 1 M to 4 M KCl at 10 °C and by 0.4% from 1 M to 4 M at 50 °C, while the RMSF of non-solvent-exposed residues increases by 12.1% and 9.2% from 1 M to 4 M at 10 °C and 50 °C, respectively. The RMSF of solvent-exposed residues increases by 20.8% from 10 °C to 50 °C in 1 M KCl and by 14.8% from 10 °C to 50 °C in 4 M KCl, while the RMSF of non-solvent-exposed residues increases by 15.5% and 12.5% from 10 °C to 50 °C at 1 M and 4 M KCl, respectively. The solvent-exposed residues have a higher percent increase in RMSF compared to the non-solvent-exposed residues when the temperature increases from 10 °C to 50 °C in both 1 M and 4 M KCl, and the non-solvent-exposed residues have a higher percent increase in RMSF compared to the solvent-exposed residues when the molarity increases from 1 M to 4 M KCl at both 10 °C and 50 °C.

The differences in the protein’s behavior at 1 M vs. 2–4 M KCl are most distinct when the SASA, RMSF, and *k*_cat_ values are all considered together. When comparing across salt concentrations, the protein in 1 M KCl has a lower SASA compared to in 2–4 M KCl across all temperatures (Figure 6b), with differences being the most distinct at the lower temperatures. Additionally, there is a general trend where the SASA decreases with increasing temperature across all molarities. When combined with the findings from the RMSF calculations (Figure 6a), the data show that higher temperature causes the protein to become less solvent exposed and more flexible while higher molarity causes the protein to become more solvent exposed and more flexible (Figure 4). The trend seen in RMSF values is similar to the trend seen in the *k*_cat_ values, where the *k*_cat_ values increase with temperature and molarity. Considering the SASA, RMSF, and *k*_cat_ values together, this suggests that the protein behaves differently in 1 M KCl than in 2–4 M KCl, especially at low temperatures (Figure 6c). At the lower temperatures (10–30 °C), the protein has similar *k*_cat_, SASA, and RMSF values in 2–4 M KCl, whereas all these values are lower in 1 M KCl, suggesting distinct enzyme behavior at 1 M KCl versus the higher molarities.

## 3. Discussion

In order to expand our understanding of polyextremophilic enzymes, on which there is a paucity of data on structure, dynamics, and function, we targeted β-galactosidase from *H. lacusprofundi* to determine the effects of a wide range of salt concentrations and temperatures. Through a series of studies from genomic sequencing and bioinformatic analysis to expression and purification, modeling and X-ray structural analysis, and steady-state kinetic analysis, a detailed view of the function of this model polyextremophilic protein is emerging. These studies have shown that the enzyme is highly acidic, with a net negative charge of −65, most of which is located on the protein surface. This remarkable characteristic is a hallmark of haloarchaeal proteins and represents a primary mechanism of adaptation to the internal salinity. Genome-wide analysis also established the likely adaptive characteristics to low temperatures, which include subtle changes to the amino acid sequence distal to the active site and key surface residues [16].

In the current study, we expanded our studies on the wild-type polyextremophilic β-galactosidase through a combination of steady-state kinetic analysis and molecular dynamics simulations, connecting both kinetic and thermodynamic properties to the predicted structural dynamics under the extreme conditions. Enzyme kinetics measured with increasing salinity showed a protein that not only can tolerate high salinity, but indeed requires it for optimal activity, as demonstrated by the lower catalytic efficiency at 1 M KCl compared to 2–4 M. Moreover, the calculated activation enthalpy and entropy were consistent with an enzyme exhibiting psychrophilic character only at the high salinity and a mesophilic enzyme at 1 M KCl. Molecular dynamics simulations under the same set of conditions showed a trend toward increased flexibility, especially at the protein surface, with both higher salt concentrations and higher temperatures. These results together underscore the synergy of action of amino acid residues distal to the active site and at the protein surface for improved catalytic properties in both of these extreme conditions (see Appendix A for the surface residues highlighted on the structure).

Molecular dynamic simulations showed that active site residues have only modest increases in flexibility with increasing salinity or temperature. The effect of salinity is similar to the trend in *K*_m_ values, albeit somewhat larger due to effects from distal regions. In a comparison of psychro-, meso-, and thermophilic orthologs, similar results with increasing temperature were reported, with the psychrophile having increased flexibility overall while the active site remained rigid [21]. Differences in activation enthalpies Δ*H*^‡^ were not associated with the active site, but were instead associated with regions of the enzyme away from the active site, analogous to the loops in domains B and C of the β-galactosidase from *H. lacusprofundi*. Better conservation is generally seen around the active site in homologous proteins. In a comparison between chitobiase from psychrophilic *Anthrobacter* sp. TAD20 and mesophilic *Escherichia coli*, the catalytic domains were well conserved [22], and the observed kinetic effects with changing temperature and salinity were not directly caused by changes in the active site.

The solvent-accessible surface area and the catalytic rate constant show a similar pattern, with a drastic increase both in the SASA and the catalytic rate constant from 1 to 2 M followed by a plateau for the higher salinities. The catalytic rates observed suggest that there is a minimum salinity required for optimal activity with little to no improvement with increased salinity above the threshold of 2 M KCl. The observed flexibility of the solvent-accessible surface area is lowest for 1 M KCl while 2–4 M KCl are very similar. This suggests the salinity effect on the catalytic rate is influenced by the “softness” as it is described by Aqvist et al., or flexibility, of the surface of the protein [21]. Our previous mutagenesis studies have shown that residues distant from the active site can have an effect on the catalytic rate of this polyextremophilic protein [15,16]. This is similar to the effects seen with temperature with cold-adapted proteins in general having more surface flexibility [23]. This trend is also demonstrated in our study, with more solvent-accessible surface area predicted at the lower temperatures (Figure 6b). 

Previous proteomic analysis has shown there is a high prevalence of acidic side chains on the surface of halophilic enzymes [4,9]. The higher prevalence of glutamic acid and aspartic acid in halophilic proteins enables them to function despite low water activity through superior water binding capability and facilitate competition with salt ions for water molecules and enhance the surrounding hydration shell [1]. The adaptive effect of a highly negative surface has been observed in high-resolution X-ray structures, in which a multi-layered hydration shell and increased flexibility through electrostatic repulsion were observed [5,24]. Haloarchaeal proteins have also evolved increased flexibility and reduced surface hydrophobicity [5], likely with the negatively charged acidic side chains also enhancing binding of hydrated cations [25].

When comparing the activation enthalpies and entropies measured for other β-galactosidases [26], the model polyextremophile studied here exhibits behavior closer to the thermo- and mesophiles at 1 M, and closer to psychrophiles at higher salinities. This suggests that the addition of salt allows the cold-active β-galactosidase to function better at lower temperatures while the functional range is diminished at lower salinities. Reduced activation enthalpies Δ*H*^‡^ seen in cold-active proteins usually occur along with more negative activation entropies Δ*S*^‡^ [27]. This feature is seen in *H. lacusprofundi* β-galactosidase, and becomes more pronounced as the salinity increases, suggesting an increase in salinity allows the protein to be more cold active [5]. A larger, more positive Δ*S*^‡^ such as that seen at 1 M KCl, suggests a more compact protein [28]. At 1 M KCl, the β-galactosidase protein has less solvent-exposed surface area and that surface is in general less flexible than at 2–4 M KCl, which correlates with the greater values of Δ*S*^‡^ seen for 1 M compared to the higher salinities. These observations suggest that the *H. lacusprofundi* β-galactosidase protein is more compact at 1 M KCl.

Future studies of β-galactosidase using Markov state models (MSMs) of the enzyme in all of the conditions would be valuable. An MSM is likely to provide insights into conformational changes associated with substrate binding, and allow us to compare predictions of *k*_off_, *k*_on_, and *K*_M_ rates with the experimental rates measured in this study. A more detailed examination of the solvation structure of ions and water seen in simulations may also reveal insights into the molecular mechanisms underlying adaptation of halophilic proteins in general. While proteomic studies have shown that an increase in the number of acidic residues on the surface can improve function in high salinity, the precise mechanisms are still not precisely established [24]. The large number of acidic residues may greatly impact the surrounding water structure, while diminishing the number of intermolecular hydrogen bonds between the surrounding water and the protein as the salt ions become hydrated [29,30,31]. The effects of temperature in this process are of interest from a biotechnological perspective.

Interactions between protein subunits in solution can also be greatly altered by salts. Electrostatic screening from salt may enhance hydrophobic interactions, causing aggregation between macromolecules [32]. As a result, multiple properties (such as the solubility, binding, stability, and crystallization) of non-halophilic proteins that have not adapted to function under high-salt conditions may be negatively affected when the proteins are introduced to such conditions [33]. The use of molecular dynamics simulations alongside MSM approaches would allow for deeper exploration of these processes, and how they relate to observed temperature- and salinity-dependent kinetic effects in halophilic enzymes.

## 4. Materials and Methods

### 4.1. Steady-State Kinetics

Kinetic experiments were performed at temperatures from 10 to 50 °C in 10 °C increments, using a Shimadzu UV-VIS 1601 spectrophotometer with a customized temperature control system designed to prevent condensation at lower temperatures using the same method described in Laye et al. [15]. A solution of 10 µg/mL of enzyme in 1–4 M KCl and 100 mM PO_4_ buffer at pH 6.5 was used for all kinetic reactions. The assay solution was preincubated at the desired temperature for 2 min before the addition of ONPG. Changes in absorption at 420 nm over 2 min and 30 s were recorded. Three different concentrations of ONPG were used: 1, 2.5, and 5 mM. All experiments were run in triplicate and values were averaged. *V*_0_, the Michaelis constant (*K*_m_), and *V*_max_ values were determined using Lineweaver–Burk plots with UV Probe ver. 4.23 software (Shimadzu). *k*_cat_ values were calculated from *V*_max_ and enzyme concentrations. The RSQ function in Microsoft Excel was used to determine *R*^2^ values of linear regression. The LINEST function in Excel was used to determine SEs for *K*_m_, *V*_max_, and *k*_cat_.

### 4.2. Bayesian Inference of Activation Enthalpies and Entropies

A Bayesian inference algorithm was implemented to estimate the values of ∆*H*^‡^ and ∆*S*^‡^ and their uncertainty, assuming the temperature-dependent *k*_cat_ measurements can be modeled by the Eyring equation: *k*_cat_ = κ (*k*_B_*T*/*h*) exp(–∆*H*^‡^/*RT* + ∆*S*^‡^/*R*), where the transmission coefficient κ is set to 1. In contrast to a simple maximum-likelihood approach (which minimizes a weighted sum of squared errors), our Bayesian approach estimates the ∆*H*^‡^ and ∆*S*^‡^ as the maximum a posteriori (MAP) of the Bayesian posterior distribution of these parameters given the experimental data *D.* Our model for the posterior probability is
*P*(∆*H*^‡^, ∆*S*^‡^, γ|*D*) ∝ *P*(*D*|∆*H*^‡^, ∆*S*^‡^, γ) *P*(∆*H*^‡^) *P*(∆*S*^‡^) *P*(γ),
where *P*(*D*|∆*H*^‡^, ∆*S*^‡^, γ) is the likelihood of observing the data given ∆*H*^‡^, ∆*S*^‡^, and γ, which is an unknown nuisance parameter describing the quality of the fit. The remaining terms are prior distributions: *P*(∆*H*^‡^) and *P*(∆*S*^‡^) are uniform priors, and *P*(γ) ~ 1/γ is the non-informative Jeffreys prior. As a model of the likelihood, we use a multi-variate Gaussian,
*P*(*D*|∆*H*^‡^, ∆*S*^‡^, γ) = Π*_i_* (2π)^−1/2^(γσ*_i_*)^−1^ exp[−(*y_i_*(∆*H*^‡^, ∆*S*^‡^) − *y_i_**)^2^/2(γσ*_i_*)^2^ ],
where *y_i_* and *y_i_** are the predicted and experimental values of *k*_cat_ at each temperature, and σ*_i_* are the reported uncertainties in the *k*_cat_ measurements. 

To find the MAP values of ∆*H*^‡^, ∆*S*^‡^, a Markov chain Monte Carlo (MCMC) algorithm was used to sample over a fixed grid of 500 ∆*H*^‡^ values ranging over 100 kJ mol^−1^, 500 ∆*S*^‡^ values ranging over 100 J K^−1^ mol^−1^, and 1000 values of γ ranging from 0.01 to 100. Values of γ were logarithmically spaced to enforce the Jeffreys prior. Proposed moves were simultaneous nearest-neighbor changes in ∆*H*^‡^, ∆*S*^‡^, and γ, with acceptance determined by the Metropolis criterion. A total of 10^7^ MCMC steps were performed, with a typical acceptance ratio of 0.80. The best-fit ∆*H*^‡^, ∆*S*^‡^ were chosen as the values that maximized the marginal distribution *P*(∆*H*^‡^, ∆*S*^‡^|*D*) = ∫*P*(∆*H*^‡^, ∆*S*^‡^, γ|*D*) *d*γ. The uncertainties in best-fit ∆*H*^‡^ and ∆*S*^‡^ were calculated as the standard deviations of marginal distributions *P*(∆*H*^‡^|*D*) and *P*(∆*S*^‡^|*D*), respectively. 

### 4.3. Molecular Modeling and Simulation

#### 4.3.1. Homology Model Construction

The MODELLER (v9.21) software package [34] in UCSF Chimera [35] was used to construct a homology model of *H. lacusprofundi* β-galactosidase from the closely related thermophilic *Thermus thermophilus* β-galactosidase structure (PDB ID: 1KWK) as a structural template, following previous similar efforts [15]. This homology model was used in all of our simulation studies.

#### 4.3.2. Homology Model Validation

The recent publication of the X-ray crystallographic structure of β-galactosidase from *H. lacusprofundi* [14] confirms the high quality of the homology model, which achieves a root-mean-squared deviation (RMSD) of 0.119 nm for the backbone alpha-carbons in domain A and 0.437 nm for all backbone alpha-carbons. The average secondary structure assignments throughout the trajectories show high agreement (the agreements range from 86% to 88%, with an average agreement of 87% across all conditions) with the secondary structure assignments of the X-ray crystallographic structure (Appendix A), with the largest disagreement (in all of the conditions) spanning only six residues (residues 473–478). In this region, our simulations display a coil, while the crystal structure contains an alpha helix (Appendix A). The remaining disagreements of secondary structure assignment are minor, often occurring for residues where the crystal structure has poor density.

#### 4.3.3. Molecular Dynamics Simulations

The protonation states of ionizable residues were determined using PROPKA [36,37], resulting in a total net charge of -68*e*. Molecular simulation topologies were constructed by solvating the homology model with TIP3P waters in a (10.98 nm)^3^ periodic box, and adding neutralizing K+ and Cl- counterions at concentrations of 1M, 2M, 3M, and 4M. The Amber14sb force field [38] was used for protein and ions. Into the solvent of this system (away from the active site) were placed three ONPG molecules, resulting in a ligand concentration of 3.628 mM. The ONPG molecular topology was constructed using the General AMBER Force Field (GAFF,) with AM1-BCC charges [39] via AmberTools18 [40]. The total number of atoms in the system was approximately 127,000. 

The enzyme was simulated as a monomer instead of its published trimeric form in an effort to closely match our kinetic experiments, which were conducted with the monomeric enzyme. Additionally, Karan et al. found no significant difference in backbone RMSD when they simulated their X-ray crystallographic structure in its monomeric and trimeric forms [14].

Simulations were performed using the GROMACS 5.1.4 software package [41]. Systems were minimized for a maximum of 50,000 steps using steepest descent minimization. Next, NPT equilibration was performed for 200 ps using a Berendsen barostat with a 1 ps time constant. NVT production runs were then performed using a stochastic (Langevin) integrator with a friction constant of 1 ps^−1^ at 300 K. The LINCS algorithm was used to constrain the hydrogen bond lengths. PME electrostatics were used with Fourier spacing 0.16 for production run. The systems were minimized and equilibrated on Temple University’s Owlsnest High-Performance and Scientific Computing Cluster. Production runs were performed on the massively parallel distributed computing platform Folding@home [42,43], with trajectory snapshots of protein and ONPG coordinates saved every 100 ps.

#### 4.3.4. Root-Mean-Square Fluctuation Calculations

Root-mean-square fluctuation (RMSF) values were calculated for each of the 20 conditions using the MDTraj Python library (McGibbon 2015). After superposition with the first frame of the trajectory, its RMSF values were calculated for every alpha-carbon. To remove time-correlation from the trajectories, trajectories were subsampled at an interval of 2τ*_c_*+ 1 ≈ 21 ns (every 210th frame), where τ*_c_* is the autocorrelation time. Additionally, the first 20 ns of each simulation were removed to ensure the sufficient equilibration. Finally, the averages of RMSF values of the trajectories were found for each condition.

#### 4.3.5. Solvent-Accessible Surface Area (SASA) Calculations

SASA values were calculated using the Shrake and Rupley algorithm from the MDTraj Python library. Default parameters (radius of 0.14 nm, and 960 sphere points) were used for the calculation, in “atom” mode. Each trajectory was superposed to the first frame, the first 20 ns of each simulation were removed, and then the SASA values of all atoms in the protein were calculated for every 21 ns. The average SASA values were calculated for each condition.

## Figures and Tables

**Figure 1 ijms-23-15620-f001:**
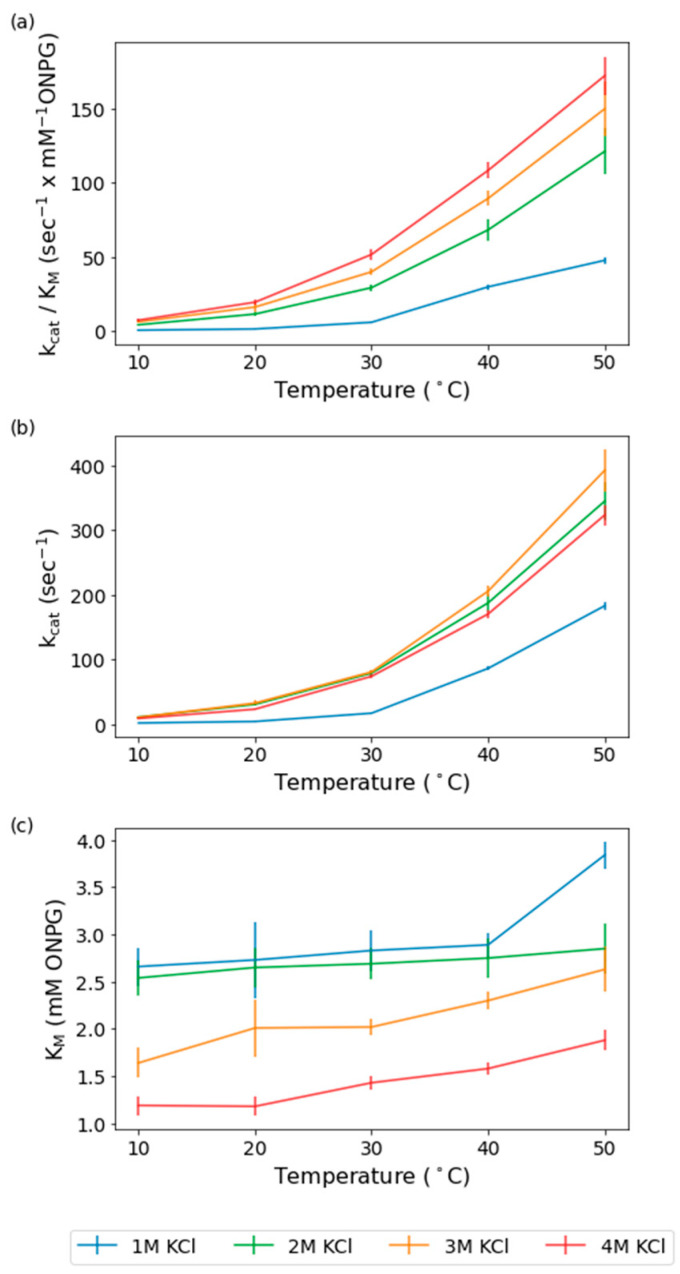
Calculated kinetic constants of wild-type for 1–4 M KCl and 10–50 °C. (**a**). Plots of the catalytic efficiency (*k*_cat_/*K*_m_, s^−1^ × mM ONPG^−1^) for the wild-type at 1 M KCl (blue), 2 M KCl (green), 3 M KCl (yellow), and 4 M KCl (red) are shown over the temperature range of 10–50 °C on the *x*-axis, with SEs shown as error bars. (**b**). Plots of the catalytic rate constant (*k*_cat_, s^−1^) for the wild-type at 1 M KCl, 2 M KCl, 3 M KCl, and 4 M KCl are shown over the temperature range of 10–50 °C on the *x*-axis, with SEs shown as error bars. (**c**). Plots of the Michaelis constant (*K*_m_, mM ONPG) for the wild-type at 1 M KCl, 2 M KCl, 3 M KCl, and 4 M KCl are shown over the temperature range of 10–50 °C on the *x*-axis, with SEs shown as error bars (*n* = 3).

**Figure 2 ijms-23-15620-f002:**
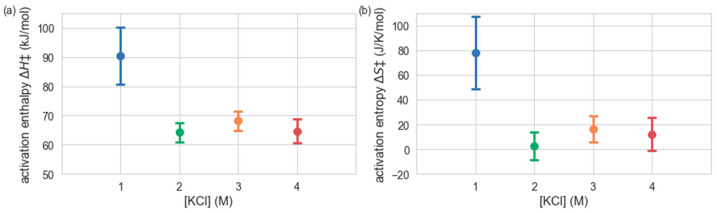
Activation enthalpies ∆*H*^‡^ (**a**) and activation entropies ∆*S*^‡^ (**b**), shown as a function of KCl concentration. Values and uncertainties were obtained by Bayesian fitting to temperature-dependent catalytic rates.

**Figure 3 ijms-23-15620-f003:**
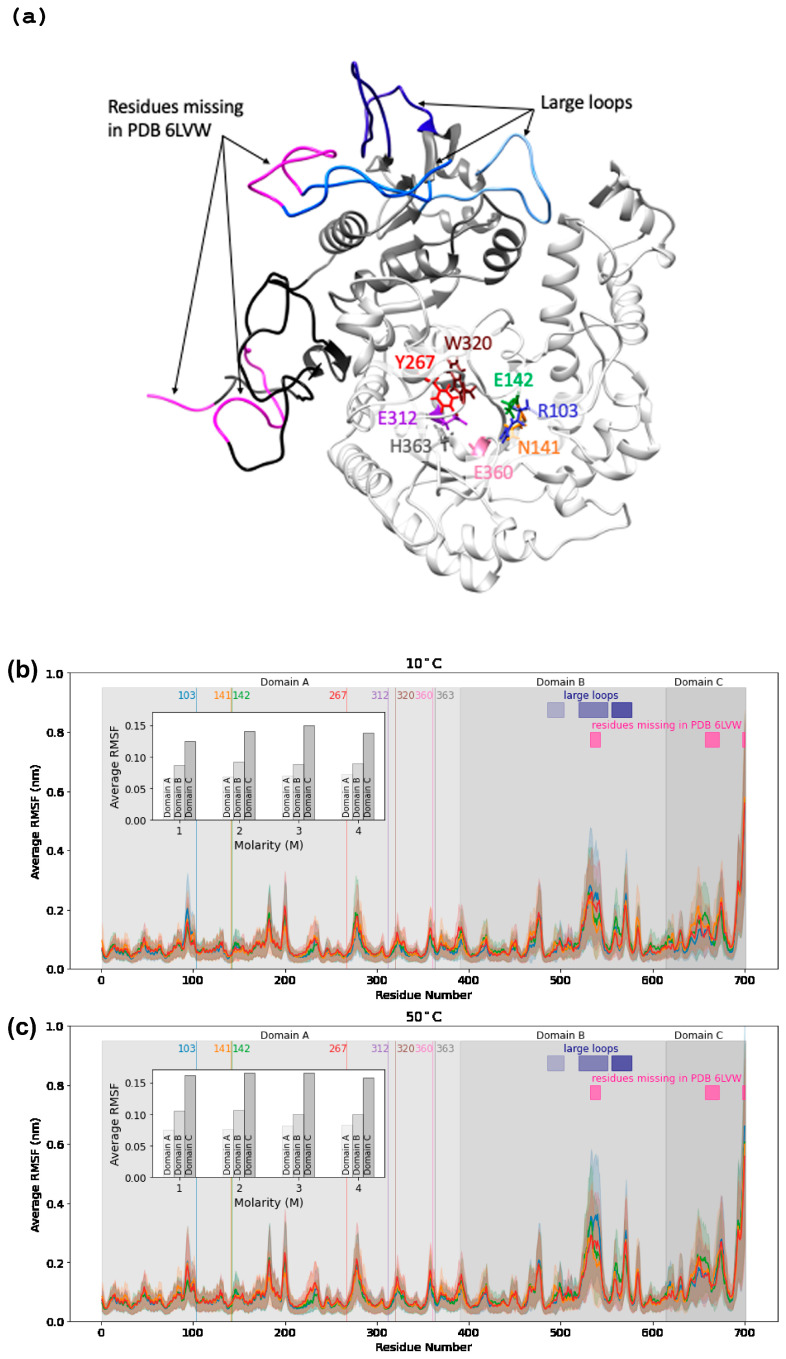
The structure of *H. lacusprofundi* β-galactosidase, with selected residues and domains labeled (**a**). RMSF profiles vs. sequence position for temperatures 10 °C (**b**) and 50 °C (**c**) with KCl molarities 1 M (blue), 2 M (green), 3 M (orange), and 4 M (red). Eight active site residues (103R, 141N, 142E, 267Y, 312E, 320W, 360E, 363H) are shown as vertical lines at their sequence positions. The three domains are shaded in light gray for domain A (residue 1–390), medium gray for domain B (residue 391–614), and dark gray for domain C (615–700). The averages of the average RMSF values per domain are shown in the inset graph on the upper left side of each graph. The sequence positions of the three large loops from domain B are shown using blue rectangles (residues 485–503, 519–551, 555–577). The sequence positions of the residues missing in the crystal structure are shown using pink rectangles (residues 532–543, 657–672, 697–700).

**Figure 4 ijms-23-15620-f004:**
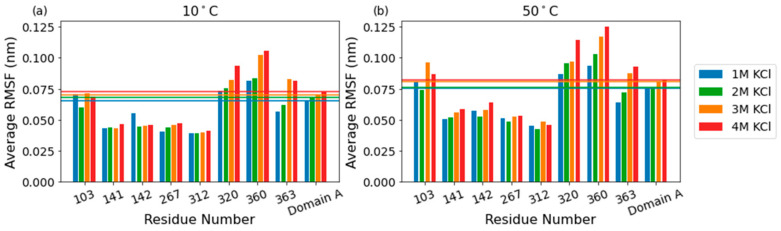
Bar graphs showing the RMSF values of the eight active site residues at 10 °C (**a**) and 50 °C (**b**). The average RMSF values per molarity for domain A are shown as horizontal lines.

**Figure 5 ijms-23-15620-f005:**
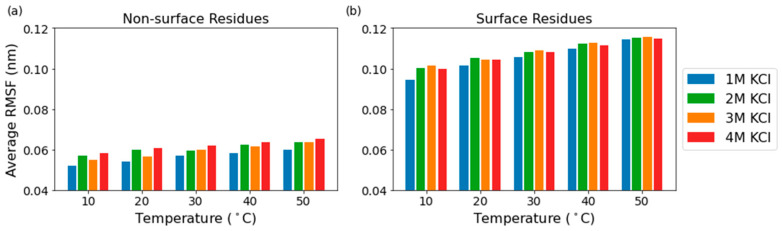
Average RMSF of the non-surface residues (**a**) and the surface residues (**b**).

**Figure 6 ijms-23-15620-f006:**
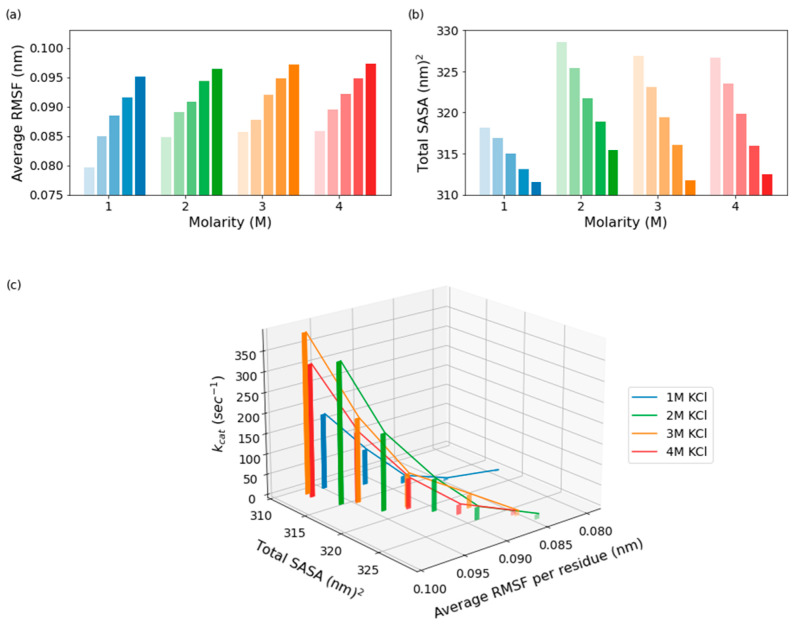
Average RMSF per residue (**a**), total SASA (**b**), and SASA vs. RMSF vs. *k*_cat_ (**c**) for five temperatures, ranging from 10 °C to 50 °C, the color of the bars becoming more saturated at the higher temperatures, and the four molarities.

## Data Availability

All analysis scripts are available at https://github.com/Solieva/BGA_manuscript (accessed on 20 May 2022). Raw trajectory data are available upon request.

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
