# Peer review of "Effects of Salinity and Temperature on the Flexibility and Function of a Polyextremophilic Enzyme"

_ijms, 2022, doi:10.3390/ijms232415620_

Round 1
Reviewer 1 Report
The paper by Laye et al is devoted to profound characterization of a polyextremophilic β-galactosidase from Halorubrum locusprofundi. In their previous work, the authors produced the recombinant enzyme and described its properties including activity dependence on temperature and salt concentration. Enzymes from extremophilic microorganisms are of high demand for biotechnology because they enable to perform catalysis in various conditions. The obtained β-galactosidase demonstrated remarkable stability in different solvents and in the presence of high salt concentration that makes it a promising candidate for industrial applications and an interesting object for study. In the current paper, using steady-state kinetics analysis, the authors revealed the complex dependence of its catalytic rate constant on salinity and temperature. To explain the observed effects, activation parameters of the enzyme at different KCl concentrations were estimated. Also, molecular dynamics simulation demonstrated increased flexibility in the regions remote from the active site at increased salinity. Taken together, obtained results clearly support conclusions explaining polyextremophilic behavior of the studied enzyme. On my opinion, the most interesting finding of the study is that at 1M salt its characteristics are closer to meso- and thermophilic homologs, while at high salinity they change to psychrophilic style. It would be interesting to support these findings by direct measurements, for example, using microcalorymetry at different salt concentrations.
The purpose of the work is clearly stated and the paper is written in a classic and convincing style. The experiments have been carried out properly. Therefore, I recommend publication in International Journal of Molecular Sciences
Reviewer 2 Report
DasSarma et al., addressed the mechanism of the polyextremophilic β-galactosidase enzyme function in the tolerance of different temperatures and high salt conditions using the steady-state kinetics and molecular dynamics simulations. That is important work. However, I can get little information about the enzyme in the manuscript, and I doubt if these evidences the work provided is enough to get the mechanism. Besides, the manuscript structure is difficult to read, thus losing readers' attention.
Reviewer 3 Report
The manuscript entitled "Effects of salinity and temperature on the flexibility and function of a polyextremophilic enzyme" from Victoria Laye et al. provides a biochemical investigation of the cold-active and salinity tolerant β-galactosidase enzyme from Halorubrum lacusprofundi.
The authors focused on the relationship between salinity/temperature and the variation of kinetics parameters obtained from well known models, and a Bayesian estimate of the thermodynamic parameters.
Moreover they try to model the protein of interest and simulate it under different conditions for providing a molecular rationale to the observed trends.
Although the used toolbox of methods is interesting for assessing the effects of both salts and temperature on this polyextremophylic enzyme, the results are in part poorly described and commented (resulting unclear) and in part potentially wrong. In the following the list of points to be adressed by the authors:
ABSTRACT
- Lines 14-15 "Km is reduced with decreasing temperatures, consistent with improved substrate binding at low temperatures". The performed experiments (Figure 1) indicate that the effect of temperature on km is milder than salinity, that abruptly decreases the km > 2 M, but there is no mention in the abstract.
- Lines 21-22: "adaptation to extreme conditions occurs via regions distant to the active site and surface accessible residues" Why is adaptation related mainly to non exposed residues? This point is not clearly presented in the discussion section.
INTRODUCTION
- Line 40-41: "An excellent source such polyextremophilic" to "An excellent source of such polyextremophilic".
- Line 45: if the temperature range of the lake was indicated correctly to be 11.5 to 18°C, it will not freeze even without salts. Fix
RESULTS
- Lines 105 - 107: the catalytic efficiency is highly influenced by salts only at higher temperatures. There seem to be a scaling problem with the data from figure 1 reported in the text: all tha data in all conditions must be normalized to the same value for allowing meaningful relative comparisons, while the authors have used different scaling factor according to different temperatures, reporting partially misleading results.
- Lines 119 - 124 and lines 137 - 143: all the values of catalytic efficiency performed at the various temperatures and salt concentrations must be scaled with the same reference. Indeed, the absolute increase of activity at 10°C with increasing salt concentration is much lower than what observed at higher temperatures, contrary to what is argued in this part of the text. Therefore, extensive editing is required here.
- Lines 131 - 137 repeat concepts from lines 119 - 124, with the same criticity.
- Lines 146-153. Figure 1 suggests that Km values lower with an increased salinity, while the increasing Km trend at increasing temperatures is similar at 1, 3 and 4 M and not related to salinity (it just started from a lower value at lower temperatures). Therefore the text must be revised here.
- Line 161: is the +DeltaS at the exponent of the "e" in the Eyring equation the activation entropy? If so, it should be concluded that it is an higher, and not a lower (as reported by the authors), activation entropy that compensate for the Kcat lowering at lower temperatures.
On the basis of these considerations, the estimated decrease in DeltaS, which is presented in the following, cannot be interpreted in the light of this formula and a different explanation should be given in this section.
- Figure 3a is poorly presented with respect to the description given in the text. The different loops and the active site should be labelled explicitly in the image to understand the connection between the image and the text.
- Lines 231 - 232: the logic of this statement is unclear.
- Lines 236-238: authors should consider that this is usual for folded globular proteins, regardless they are cold-active or not.
- Lines 253-255: not clear.
- Lines 257-258: simplify and avoid to repeat RMSF multiple times.
- Line 262: you are making predictions from simulations, so I suggest to replace "it is clear" with "this suggests ..." or something similar.
- Lins 264 - 266: ok, but from what I can see in figure 1, the major effect of salts is not related to the change in the kcat, which increases mainly upon temperature increase, but to the Km.
DISCUSSION
- Lines 294 - 296: residues can be both distal and on the surface. Please, you may want to explicitly cite MD simulation data on the difference between flexibility of surface residues that are distal or proximal to the active site.
- Lines 298 - 303: these two statements are not understandable, please rewrite them.
- Lines 303 - 305: which regions? Please indicate at least in the text their precise position in the sequence and in the structure.
- Lines 307 - 309 : I don't think changes of the active site related to any adaptation are common and it is not the focus of the cited research.
- Lines 312-314: this claim is repeated multiple times across the manuscript.
- Lines 314-316: bad sentence construction. The claim of the authors is not clear. What is the "softness" of the protein surface? Why is it associated with superior flexibility?
- Lines 320 - 321: this is in contrast with the result you presented in lines 245 - 246.
- Lines 337-338: are the authors sure that activation entropy is negative?
Look at this ref.: https://doi.org/10.3389/fmicb.2016.01408.
It seems that what is negative is the activation entropy difference between a mesophilic and a psychrophilic enzyme, but the absolute activation entropy is positive.
- Line 340: I don't think a superior entropic contribution lead to a more compact protein. It usually leads to superior comformational variation. This part of the discussion should be rewritten to provide another explanation to the collected data
- Line 360: this is true if a folded protein has a relatively big fraction of hydrophobic residues exposed, while in most of the cases the reported scenario of high salts is stabilizing (even if inihibiting the activity) for most enzymes. What is the situation of exposed residues for the beta-galactosidase under study?
- Lines 361-363 are not clear
- Lines 412 - 416: was the oligomerization state of the template structure considered? Is the same of the experimentally solved structure?
If there are discrepacy with the oligomerization, the authors may perform at least a part of the simulations with the real structure to confirm the correlation of their results with that obtained from the homology model
- Line 439: probable error in formatting the sub-paragraphs?
Round 2
Reviewer 2 Report
Accept
Reviewer 3 Report
The detailed explanations of the authors and their modifications to the manuscript are clear and convincing